# PUTNAMBENCH: Evaluating Neural Theorem-Provers on the Putnam Mathematical Competition

**George Tsoukalas**
UT Austin

**Jasper Lee**
UT Austin

**John Jennings**
UT Austin

**Jimmy Xin**
UT Austin

**Michelle Ding**
UT Austin

**Michael Jennings**
UT Austin

**Amitayush Thakur**
UT Austin

**Swarat Chaudhuri**
UT Austin

## Abstract

We present PUTNAMBENCH, a new multi-language benchmark for evaluating the ability of neural theorem-provers to solve competition mathematics problems. PUTNAMBENCH consists of 1692 hand-constructed formalizations of 640 theorems sourced from the William Lowell Putnam Mathematical Competition, the premier undergraduate-level mathematics competition in North America. All the problems have formalizations in Lean 4 and Isabelle; a substantial subset also has Coq formalizations. PUTNAMBENCH requires significant problem-solving ability and proficiency in a broad range of topics taught in undergraduate mathematics courses. We use PUTNAMBENCH to evaluate several established neural and symbolic theorem-provers. These approaches can only solve a handful of the PUTNAMBENCH problems, establishing the benchmark as a difficult open challenge for research on neural theorem-proving. PUTNAMBENCH is available at https://github.com/trishullab/PutnamBench.

## 1 Introduction

Automating mathematical reasoning is a longstanding goal in artificial intelligence (Newell et al., 1957). A prominent line of work on the problem (Li et al., 2024) uses neural models to direct theorem-proving in formal frameworks like Lean 4 (Moura & Ullrich, 2021), Isabelle (Wenzel et al., 2008), and Coq (The Coq Development Team, 2023). These frameworks can "execute" proofs like code and offer execution feedback, which simplifies the search for correct proofs.

The design of quality benchmarks is a key challenge in this research area. The two most prominent competition-based benchmarks for neural theorem-proving are MINIF2F (Zheng et al., 2021) and FIMO (Liu et al., 2023). The former formalizes a mix of problems from high-school level courses and mathematics competitions such as AIME, AMC, and IMO; the latter consists of a collection of IMO problems. Both benchmarks have limitations. For example, MINIF2F contains many problems that can be immediately solved using an SMT solver, and FIMO only targets the Lean 3 framework, which is no longer actively maintained.

More generally, as large language models (LLMs) grow in importance as a tool for neural theorem-proving (Li et al., 2024), preventing leakage between pretraining sets and evaluation sets is more important than ever. This makes the continued supply of new benchmarks an important goal.

In this paper, we respond to this challenge with PUTNAMBENCH, a new hand-curated, multi-langauge benchmark for neural theorem-provers. PUTNAMBENCH includes 1692 formalizations of 640 problems from the William Lowell Putnam Mathematical Competition, the premier college-level

mathematics competition in North America.[*] All our problems have Lean 4 (Moura & Ullrich, 2021) and Isabelle (Wenzel et al., 2008) formalizations; a substantial fraction have formalizations in Coq (The Coq Development Team, 2023) as well. The formalizations are all manually constructed and have been carefully debugged. The benchmark also includes the original English-language problem statements with permission from the Mathematical Association of America, which organizes the Putnam competition.

One key benefit of PUTNAMBENCH is that Putnam competition problems require a broad base of mathematical knowledge and skills. Because they target undergraduate students, they cover topics such as analysis and abstract algebra that do not appear in the International Mathematical Olympiad (IMO). At the same time, success in the two competitions is correlated — top performers on the Putnam competition are often former IMO medalists as well. Hence, PUTNAMBENCH is well-aligned with the IMO Grand Challenge (Challenge, 2019) and the AI Mathematical Olympiad (Prize, 2023), the latter of which offers a $10M prize fund for developing a system that can win a gold medal at the IMO.

Another advantage is that PUTNAMBENCH supports multiple proof assistants. Lean 4, Coq, and Isabelle are currently the three most popular formal proof languages. However, theorem-proving benchmarks typically only contain problems in a strict subset of these languages — for example, MINIF2F (Zheng et al., 2021) does not include Coq problems, and FIMO (Liu et al., 2023) only targets Lean. PUTNAMBENCH is the first mathematics-competition benchmark to include problems in all three languages.

We use PUTNAMBENCH to evaluate several neural and symbolic approaches: Draft-Sketch-Prove (Jiang et al., 2022b), COPRA (Thakur et al., 2024), GPT-4, Sledgehammer (Paulson & Blanchette, 2015), and Coqhammer (Czajka & Kaliszyk, 2018). Collectively, these methods can only solve a handful of the PUTNAMBENCH problems, establishing PUTNAMBENCH as a hard open challenge for the neural theorem-proving community.

## 2 Background

**Formal Theorem-Proving.** Formal proof frameworks like Lean 4 (Moura & Ullrich, 2021), Coq (The Coq Development Team, 2023), and Isabelle (Wenzel et al., 2008) allow users to write machine-verifiable proofs of mathematical theorems. To create such a proof, one first uses a framework-specific language to formally state the target theorem. The mathematical objects referenced in the theorem can be imported from an existing repository or defined by the user. During the proof process, the proof framework maintains a *state* that includes information about the parts of the proof that remain to be completed. One can change this state by executing a *proof step*. The user's goal is to write a sequence of proof steps (in the framework's language) that changes the proof state to a special state "QED" in which there are no unmet proof obligations.

Figure 1 illustrates a theorem and proof in the Lean 4 framework.

```
theorem putnam_1988_b1 :
∀ a ≥ 2, ∀ b ≥ 2, ∃ x y z : ℤ,
x > 0 ∧ y > 0 ∧ z > 0 ∧
a * b = x * y + x * z + y * z + 1 := by
    intro a ha b hb
    use a - 1, b - 1, 1
    constructor
    linarith
    constructor
    linarith
    constructor
    linarith
    ring
```

Figure 1: A formalization of Putnam 1988 B1 in Lean 4, which asserts that for all integers $a, b \geq 2$, there are positive integers $x, y, z$ such that $ab = xy + xz + yz + 1$. The formal proof begins by introducing all relevant variables and hypotheses with `intro`, then indicating the choice of $x, y, z$ with `use`, and afterwards proving all goals using the automated tactics `linarith` and `ring`. This proof was discovered through a few-shot invocation of GPT-4.

**The Putnam Competition.** The William Lowell Putnam Mathematical (Competition, 2024), organized by the Mathematical Association of America (MAA), is the premier collegiate mathematics competition in North America. Thousands of undergraduate students from universities across the United States and Canada take the exam each year. The competition comprises two 3-hour-long sessions of six problems each, presented in approximately ascending order of difficulty within each

---

[*]PUTNAMBENCH is available at https://github.com/trishullab/PutnamBench.

| Benchmark | # | Natural Language | Lean | Isabelle | Coq | Factored Solution |
|---|---|---|---|---|---|---|
| MINIF2F | 488 | ✓ | ✓[†] | ✓ | | |
| PROOFNET | 371 | ✓ | ✓[†] | | | N/A |
| FIMO | 149 | ✓ | ✓[†] | | | |
| PUTNAMBENCH | 640 | ✓ | ✓ | ✓ | ✓ | ✓ |

Table 1: Comparison of existing formal theorem proving evaluation benchmarks. PUTNAMBENCH exceeds prior benchmarks by providing support for all of Lean 4, Isabelle, and Coq, on a set of difficult competition problems using undergraduate-level mathematics. For problems requiring a numerical solution in addition to a proof, we factor the solution out of the theorem statement.

session. While some problems require competitors to furnish a concrete solution (such as a number, a set, or the truth value of a given statement), all problems require a natural-language proof of correctness. The contest draws from a wide variety of topics in the undergraduate curriculum, often using instances of ideas from research-level mathematics.

## 3 PUTNAMBENCH

PUTNAMBENCH is a multi-language evaluation benchmark consisting of formalized problems from the Putnam competition. PUTNAMBENCH is a manually produced benchmark, including 640 formalizations in Lean 4 and Isabelle, and 412 formalizations in Coq. In aggregate, PUTNAMBENCH contains 1692 formalizations of Putnam competition problems. We also incorporate the informal statements and numerical solutions where applicable.

Now we elaborate on the main features of PUTNAMBENCH.

**Diversity and Breadth.** Compared to MINIF2F (Zheng et al., 2021) and FIMO (Liu et al., 2023), which generally rely on high-school mathematics, PUTNAMBENCH incorporates a wider variety of problems which require definitions of the standard undergraduate mathematics curriculum. The PROOFNET benchmark (Azerbayev et al., 2023) also sources problems from the undergraduate curriculum, but these problems are generally from standard textbooks as opposed to mathematical competitions. Putnam problems often require definitions from multiple fields, which standard textbooks do not necessarily

| Category | Total Quantity |
|---|---|
| Algebra | 253 |
| Analysis | 226 |
| Number Theory | 107 |
| Geometry | 68 |
| Linear Algebra | 51 |
| Abstract Algebra | 28 |
| Combinatorics | 26 |
| Probability | 9 |
| Set Theory | 8 |

Table 2: Quantity by domain of PUTNAMBENCH problems. Our formalizations generally reflect the variety of Putnam problems, though we can only formalize few geometry and probability problems due to limited support for these topics in the respective mathematical libraries.

target. Formalizations in PUTNAMBENCH include concepts from a wide range of mathematical fields, including: (i) *Analysis*: Limits, integrals, derivatives, continuity; (ii) *Linear Algebra*: Matrices, determinants, fields; (iii) *Abstract Algebra*: Rings, groups, magmas, permutations; (iv) *Algebra*: Polynomials, inequalities, algebraic expressions; (v) *Number Theory*: Primes, irrationality, base representations, divisors, palindromes; (vi) *Geometry*: Polygons, point sets, line intersections, Euclidean distance; (vii) *Set Theory & Combinatorics*: Countability, power sets, discrete structures, games.

**Multiple Languages.** PUTNAMBENCH contains formalizations of Putnam problems in Lean 4, Isabelle, and Coq. The formalizations also include concepts defined in each proof assistant's mathematical repositories — notably, Mathlib, the HOL standard library, and Coquelicot (among various Coq repositories). To the best of our knowledge, PUTNAMBENCH is the first undergraduate-level competition benchmark for each of these languages. Furthermore, we are the first to produce a human mathematics competition-style evaluation benchmark for Coq.

We hope that this contribution can enable Coq practitioners access to the rapidly-growing field of machine learning for mathematics.

Generally, the formalizations of the problems are aligned in their structure, including hypothesis naming and framing. Differences may arise according to the underlying foundations of each language.

We also note that the pre-defined mathematical theory in each language differs, which can sometimes lead to difficulties formalizing certain problems.

Compared to the prior benchmarks MINIF2F, FIMO, and PROOFNET, PUTNAMBENCH is the first to support Lean 4 on initial release [†].

**Factored Solutions.** Roughly 60% of Putnam problems, in their natural language form, require exhibiting a (closed-form) solution along with a proof of its correctness. Such problems do not assert propositions, and hence are not immediately formalizable as they are not directly the statement of a theorem. Prior benchmarks such as MINIF2F (Zheng et al., 2021) sidestep this issue by rewording the problem statement to ask for a proof that the solution satisfies the constraints of the problem. However, this reduction diminishes the overall difficulty of the problem, as producing a solution can constitute the majority of the difficulty. To address this issue, we factor out solutions of such problems from the formalized theorem statement. We include an example in Figure 2. In this way, we provide two tasks for neural theorem proving:

- **Task 1:** Given the theorem statement, first identify the (closed-form) solution, and then provide a proof of correctness by rewriting the solution into the theorem statement.

- **Task 2:** Given the theorem statement and solution, produce a proof of its correctness. This task aligns with the current benchmarks.

We note that the process of producing the numerical solution may be highly correlated with the proof of its correctness. In this way, our formalizations can reflect the true difficulty of the informal problem statement.

**Formalization effort and challenges.** We hand-crafted our benchmark over the course of several months as a team of two doctoral and five undergraduate students with prior experience in university mathematics, computer science, and formal proof assistants. We found that the average time-to-formalize a single problem in one language was roughly 25 minutes. Each formalization was verified by a second person at least once, and we measured that the verification of a single formalization took between 10 minutes, on average. We acknowledge that the time-to-formalize we report is higher than that of MINIF2F; we believe this is largely due to the increased complexity of the Putnam problems, which oftentimes require definitions we must locate in each language's respective mathematical libraries.

We first produced formalizations in Lean 4, and then proceeded with our formalization effort in Isabelle and then Coq. Due to differences in the underlying foundations of each language, we found that formalizations in one language sometimes do not directly transfer to another; for example, Isabelle does not have a subtyping mechanism, which we made extensive use of in Lean 4. Formalizations in Coq rely on a number of mathematics repositories. Predominantly, we rely

> **Putnam 2008 B5.** Find all continuously differentiable functions $f : \mathbb{R} \to \mathbb{R}$ such that for every rational number $q$, the number $f(q)$ is rational and has the same denominator as $q$.

```
abbrev solution : Set (ℝ → ℝ) :=
    {fun (x : ℝ) => x + n | n : ℤ} ∪
    {fun (x : ℝ) => -x + n | n : ℤ}
theorem putnam_2008_b5
(fqsat : (ℝ → ℝ) → ℚ → Prop)
(hfqsat : ∀ f q, fqsat f q ↔
    ContDiff ℝ 1 f ∧
    (∃ p : ℚ, p = f q ∧ p.den = q.den)) :
∀ f : (ℝ → ℝ), (∀ q : ℚ, fqsat f q)
↔ f ∈ solution :=
```

Figure 2: A formalization of Putnam 2008 B5 in Lean 4. As the problem requires exhibiting the set of functions $f$ satisfying the specified conditions, it is not directly the statement of a theorem. We formalize the problem by instantiating a variable "solution" outside of the theorem statement. In this way, a model can either provide its own candidate, or use the correct solution we provide and attempt to produce a proof of correctness. Benchmarks such as MINIF2F and FIMO only include formalizations with the solution written into the theorem statement.

---

[†]MINIF2F, FIMO, and PROOFNET were originally released using Lean 3, and MINIF2F and FIMO have been updated to include Lean 4 formalizations following community efforts. (Azerbayev et al., 2023; Vishwakarma et al., 2024). To the best of our knowledge, no open-sourced Lean 4 version of FIMO currently exists.

on MathComp and MathComp-Analysis (Mathcomp, 2015; mathcomp-analysis), but also make us of Stdlib, Stdpp, Coquelicot, GeoCoq, and Coqtail (Coquelicot, 2015; GeoCoq, 2015; Allais et al.).

Some problems are not naturally amenable to formalization — for example, we found that while formalizing problems involving probabilities is possible, such formalizations often require heavy probability theory. Similarly, support for problems involving Euclidean geometry varies across languages; in particular, Lean 4 does not yet have a sufficiently extensive library to make most geometry problems formalizable. By contrast, Coq has an extensive geometry repository called GeoCoq, which we utilize for our Coq formalizations.

**Dataset Contamination.** Our benchmark is unique compared to informal benchmarks such as MATH (Hendrycks et al., 2021) and GSM8K (Cobbe et al., 2021) in the sense that the target output *has never been produced*, hence avoiding direct contamination. To the best of our knowledge, we are the first to provide formalizations of a large collection of Putnam problems in any of Lean, Isabelle, and Coq. Since writing a formal proof requires the formal theorem statement, it is highly unlikely any possible formal proof has been written for any of our problems. We performed a thorough investigation of formal mathematics repositories for each language for confirmation, finding no aligned theorems and proofs from the Putnam Competition. We do not include any of the formal proofs in our benchmark.

Furthermore, any proofs found by automated methods in our evaluations are not included and are only mentioned in this article. Indirect contamination can occur through transfer from training on the informal proofs, though producing proofs in formal proof environments still presents a major difficulty for all current neural methods, as we find in Section 4.

```
(a) theorem putnam_2006_b2
(n : ℕ)
(npos : n > 0)
(X : Finset ℝ)
(hXcard : X.card = n)
: (∃ S ⊆ X, S ≠ ∅ ∧ ∃ m : ℤ,
    |m + Σ s in S, s| ≤ 1 / (n + 1))
```

```
(b) theorem putnam_2006_b2:
fixes n :: nat
and X :: "real set"
assumes npos: "n > 0"
and hXcard: "finite X ∧ card X = n"
shows "∃ S ⊆ X. (S ≠ {}) ∧ (∃ m :: int.
    ¦m + (Σ s ∈ S. s)¦ ≤ 1 / (n + 1))"
```

```
(c) Theorem putnam_2006_b2
(n : nat)
(hn : gt n 0)
(X : seq R)
(hX : uniq X /\ size X = n)
: exists S : seq R,
    subseq S X /\
    size S <> 0%nat /\
    exists m : int,
    `|m%:~R + \sum_(s <- S) s| <= 1 / (n%:R + 1).
```

Figure 3: Formalizations of Putnam 2006 B2 in (a) Lean 4, (b) Isabelle, (c) Coq. Putnam 2006 B2 asserts that given a finite subset $X \subseteq \mathbb{R}$ with $|X| = n > 0$, there is a nonempty subset $S \subseteq X$ and an $m \in \mathbb{Z}$ such that $|m + \sum_{s \in S} s| \leq \frac{1}{n+1}$.

**Licensing and Rules of Engagement.** PUTNAMBENCH is available under an Apache 2.0 license for Lean 4 and Isabelle, and under an MIT license for Coq. We align the licenses with those of the repositories we use for each language. With permission from the MAA, we include the informal statements as sourced from the competition (Alexanderson et al., 1985; Kedlaya et al., 2002, 2020). We host a public leaderboard at https://trishullab.github.io/PutnamBench/ and will readily accept evaluation results from future works.

## 4    Experimental Evaluation

To understand the challenges that PUTNAMBENCH poses for state-of-the-art theorem-proving approaches, we attempt to solve its problems using a suite of such approaches. Given the relative lack of tailored systems for multi-language theorem-proving, we run evaluations for each language separately. Any method that is evaluated on multiple languages is based on off-the-shelf foundation models.

| PUTNAMBENCH: Lean | |
| --- | --- |
| Method | Success Rate |
| GPT-4 | 1/640 |
| COPRA | 1/640 |
| ReProver (+r) | 0/640 |
| ReProver (−r) | 0/640 |

| PUTNAMBENCH: Isabelle | |
| --- | --- |
| Method | Success Rate |
| GPT-4 | 1/640 |
| DSP | 4/640 |
| Sledgehammer | 3/640 |

| PUTNAMBENCH: Coq | |
| --- | --- |
| Method | Success Rate |
| GPT-4 | 1/412 |
| COPRA | 1/412 |
| Tactician | 0/412 |
| CoqHammer | 0/412 |

Table 3: Results of evaluations on PUTNAMBENCH in each language. We find that all tested methodologies perform poorly, solving at most a handful of problems. Notably, the only problem solved in both Lean and Coq is Putnam 1988 B1, which is not solved by any method in Isabelle. ReProver, our finetuned baseline for Lean, is unable to solve any problems with or without retrieval. Symbolic automation proves to be powerful in Isabelle, with Sledgehammer solving the most problems than GPT4 alone. DSP generates four successful proofs, two of which cannot be generated by Sledgehammer alone.

**Metrics.** Our evaluation is based on the $pass@n$ (Lample et al., 2022) metric. This metric measures a prover's ability to produce a successful proof, as determined by the formal proof environment, given a budget of $n$ *proof attempts*. In search-based methods (Thakur et al., 2024), each proof attempt involves a distinct search that can query a neural model multiple times.

**Models.** For each of the languages, we perform evaluations using GPT-4 (OpenAI, 2023) [‡], a highly capable foundation model. We run evaluations using in-context learning, appending several examples of successful proofs of simple theorems in each language. For evaluations with Lean 4 approaches, we note that many approaches have targeted Lean 3, which is not backward-compatible and no longer actively maintained. We evaluate COPRA (Thakur et al., 2024) on PUTNAMBENCH, modifying the prompt examples of COPRA to enable search in Lean 4. Furthermore, we run evaluations LeanDojo's retrieval-augmented prover REPROVER, a finetuned model designed to utilize incorporate retrieved lemmas as part of the proof search. We also include evaluations with the retrieval component held out.

For our Isabelle experiments, we run evaluations of Draft, Sketch, and Prove (DSP) (Jiang et al., 2022b) using GPT-4 as the underlying foundation model, noting that many further works for theorem-proving in Isabelle have extended on the DSP pipeline as we mention in Section 5. We also run evaluations using stand-alone invocations to Sledgehammer, a powerful symbolic automation tool in Isabelle that relies on calls to external SMT solvers.

As for our Coq experiments, prior neural approaches for Coq have mostly targeted software verification tasks, as opposed to competition mathematics. As a result, our Coq experiments use COPRA, which also supports theorem-proving in Coq. We evaluate using the Tactician (Blaauwbroek et al., 2020) platform with the locality sensitive hashing model configuration. We also run evaluations using CoqHammer (Czajka & Kaliszyk, 2018), a tool similar to Isabelle's Sledgehammer, which makes calls to external constraint solvers.

### 4.1 Results

**Lean 4.** We prompt GPT-4 in a $pass@10$, setting temperature $T = 0.7$ and using several examples of simple theorems and proofs, to generate a proof for each problem. The result of this experiment yields a single successful proof across all 640 Lean formalizations. The problem (Putnam 1988 B1) and the generated proof are given in Figure 1. In particular, Putnam 1988 B1 is solved on the first of 10 attempts. An example of a failure mode of GPT-4 is given in Figure 18.

We also run evaluations with COPRA, using their default hyperparameters for search, performing a $pass@1$, and allowing 60 queries to GPT-4. However, since COPRA was originally designed for interaction with Lean 3, we make a small modification to its system prompt to enable search in Lean 4. The result of the step-wise proof search over all Lean 4 formalizations yields a correct proof to one problem (1988 B1). We find that backtracking in the search was not required for this proof, which was 10 lines long and was found at the 10th query. It is possible that affording COPRA further

---

[‡]We use GPT-4o for all our evaluations

queries to GPT-4 can yield more successful proofs, though it is not yet feasible to perform such an experiment due to the cost of queries to GPT-4.

We found that, by default, GPT-4 produces proofs using Lean 3 syntax, which is not compatible with Lean 4. Even when directed to produce outputs in Lean 4, GPT-4 typically continues to produce outputs in Lean 3. Our prompt, which we include in Figure 16, elucidates some design differences in Lean 4 to better enforce compliance with the Lean 4 syntax. However, we noticed many examples where GPT-4 continues to output terms in Lean 3 syntax. One such example is given in Figure 17.

We run REPROVER using the standard search parameters used in LeanDojo (Yang et al., 2023). Our evaluation yields no successfully proven problems, with and without the inclusion of the retrieval module. We believe that Putnam 1988 B1, which the other methods solve, is not solved by REPROVER as it requires an understanding that the choice of $x, y, z = 1, a - 1, b - 1$ will eventually satisfy the conditions of the goal after simplification. Smaller models, like the one driving REPROVER's search, may not be as readily capable of such understanding.

**Isabelle.** We run GPT-4 using the same configuration, with modified prompts for Isabelle, on our Isabelle formalizations. We find that GPT-4 can produce a single successful proof to Putnam 1986 B1, a geometric problem stated algebraically. We include the statement and its proof as generated by GPT-4 in Figure 19.

DSP represents a neurosymbolic methodology which has seen significant application for theorem-proving in MINIF2F. We run DSP with $pass@10$, using temperature $T = 0.1$ and GPT-4 as the underlying language model. Our evaluation yields four successful proofs: of Putnam 2001 A1 and 1971 B1, two problems involving magmas (sets with a binary operation), one of Putnam 1995 A1, a problem involving a closed-under-multiplication subset of the reals, and Putnam 1986 B1. In particular, Putnam 1995 A1 and 1986 B1 cannot be solved by Sledgehammer alone. The generated proof of Putnam 1995 A1 is included in Figure 4.

We run a baseline using Sledgehammer, a powerful automation tool in Isabelle which makes calls to external SMT solvers to prove a given goal. With a set timeout of $t = 120$ seconds, we run Sledgehammer on each Isabelle formalization. The result of this evaluation is 3 successfully proven problems: Putnam 1971 B1, 2001 A1, and 2012 A2. Notably, all of these problems are statements about sets with binary operations. We include the statements of 1971 B1 and 2012 A2 in Figure 22.

**Coq.** We run GPT-4 with a Coq-based prompt on our Coq formalizations using the same configuration as in Lean and Isabelle. The result of the experiment is 1 solved problem, namely Putnam 1988 B1, which was also solved in Lean 4. The proof, which we include in Figure 14, generally follows the same structure as the proof in Lean.

An evaluation with COPRA, in a $pass@1$-with-60-queries and $T = 0.0$ also yields a successful proof only for Putnam 1988 B1 which we include in Figure 14. In this case, backtracking was crucial for proof search on this problem. The crucial step in 1988 B1 is the choice of $x, y, z$ once $a$ and $b$ have been introduced. Initially, CO-PRA predicts the erroneous choice $x, y, z = 1, 1, ab - 1$ and eventually reverts this choice using backtracking. Afterwards, COPRA predicts a correct choice $x, y, z = 1, a - 1, b - 1$ and proceeds with the proof.

> **Putnam 2001 A1.** Consider a set $S$ and a binary operation $\star$, i.e., for each $a, b \in S$, $a \star b \in S$. Assume $(a \star b) \star a = b$ for all $a, b \in S$. Prove that $a \star (b \star a) = b$ for all $a, b \in S$.

```
theorem putnam_2001_a1:
  fixes op :: "'a ⇒ 'a ⇒ 'a"
  assumes hop : "∀a b :: 'a.
    op (op a b) a = b"
  shows "∀a b :: 'a. op a (op b a) = b"
proof -
  {
    fix a b :: 'a
    have "op (op a (op b a)) a = op b a" using
    hop by simp
    then have "op a (op b a) = b" using hop by
    metis
  }
  then show ?thesis by simp
qed
```

Figure 4: A formalization of Putnam 2001 A1 in Isabelle and the corresponding proof discovered by our evaluation with DSP. Sledgehammer alone can also produce a successful proof to this theorem.

We run Tactician using the locality sensitive hashing model with a timeout of $t = 600s$ per problem. Our evaluation yields no successfully proven problems. While showing favorable performance on theorems drawn from Coq's standard library (Zhang et al., 2021), such methodologies do not as of yet scale to challenging olympiad-style problems.

We run CoqHammer with 8 parallel threads using an ATP timeout of 100 seconds, proof reconstruction timeout of 15 seconds, and sauto timeout of 5 seconds, for a total of 120 seconds allocated for each formalization. The evaluation yields no successful proofs — indicating that symbolic tools in Coq are not yet capable of handling PUTNAMBENCH problems. It is not surprising that CoqHammer does not match the performance of Sledgehammer even though they rely on the same external solvers. The underlying logical system of Coq is more complex than that of Isabelle and is hence less amenable to automation.

## 4.2 General Analysis

Aggregating over all experiments performed in all languages, we find that a total of 6 problems in PUTNAMBENCH are successfully proven. A majority of these come from evaluations in Isabelle, particularly with strong contributions from Sledgehammer. Sledgehammer can solve all three problems involving magmas which appear in our benchmark but fails to produce successful proofs for any other formalization. DSP solves an additional two problems and relies heavily on Sledgehammer to fill in the proofs of intermediate steps. The single problem solved in Lean and Coq also makes use of automated tactics like `linarith` and `lia`, and requires only a single crucial step.

Hence, we find that a few PUTNAMBENCH problems are not entirely intractable using current methods. However, anecdotally, these problems are among the easiest ever included in the Putnam competition. All admit a very short natural language proof and do not require reasoning about particularly complicated objects. We believe that significant advancements in automated mathematical reasoning are required to make progress on PUTNAMBENCH.

## 5 Related Work

**Formal Benchmarks.** Several evaluation benchmarks for formal mathematics have been developed in recent years. MINIF2F (Zheng et al., 2021) is a formal-to-formal benchmark of competition problems, sourced from high school competitions such as the AMC, AIME, and IMO. MINIF2F is a multi-language benchmark, comprising of 488 problems each formalized in Lean 3, Metamath, Isabelle and HOL Light. We chose not to include formalizations in Metamath and HOL Light as they have not been the focus of attention for neural theorem-proving. A similar competition-style benchmark is FIMO (Liu et al., 2023), which contains 149 Lean 3 formalizations of IMO shortlist problems produced using a back-translation procedure with GPT-4. The automatically-generated formalizations are then manually verified. Both benchmarks are designed to measure *certifying* the solution to the informal problem statement when one exists. Compfiles (2024) is a collection of 171 Lean 4 formalizations of competition problems, predominantly from the IMO and USAMO, often accompanied by a formal proof, which has not seen use in benchmarking automated theorem-provers. ProofNet (Azerbayev et al., 2023) introduced a benchmark of 371 exercises, formalized in Lean 3, from standard textbooks in the undergraduate mathematics curriculum. While largely not competition-based, problems in ProofNet draw from a broader library of concepts than miniF2F and FIMO, which rely only on high-school mathematics. LeanDojo (Yang et al., 2023) introduces a dataset of formal mathematics and proofs derived from Lean's mathlib library (mathlib Community, 2020), and trains a retrieval-augmented model towards generating proofs on their held-out test set. ProverBot9001 (Sanchez-Stern et al., 2020) introduced a dataset for theorems and proofs written in Coq derived from CompCert (Leroy, 2009), a formally verified C compiler. PISA (Jiang et al., 2021) is a dataset derived from Isabelle's Archive of Formal Proofs (AFP), which contains theorems and proofs from general mathematics as opposed to specifically competition problems.

**Informal Benchmarks.** There are also several popular benchmarks for informal (natural-language) mathematical reasoning. MATH (Hendrycks et al., 2021) is a collection of 12,500 mathematics problems, in natural language only, sourced from various high school competitions additionally supplied with step-by-step informal proofs. GSM8K (Cobbe et al., 2021) is a collection of 8,500 grade school mathematics problems, intended to benchmark natural language reasoning for mathematics-style problems. While benefiting from the abundance of natural language data, these benchmarks fall

short, since in natural language, there is no automatic mechanism for certifiable verification of the reasoning path which yielded the numerical answer. For this reason, metrics for success on these benchmarks usually rely on exact-answer match, because verifying reasoning paths is imprecise and is best done by human experts. By contrast, theorem proving in formal proof assistants comes with a high-confidence signal for correctness of the reasoning path, or *proof*, of a theorem.

**Methods for Formal Theorem-Proving.** Significant effort has been spent on developing automatic theorem-provers for formal mathematics (Li et al., 2024). Most recent efforts train a neural module to perform proof-step prediction, which is then wrapped in a search mechanism to locate a valid proof. GPT-$f$ (Polu & Sutskever, 2020) trains a transformer-based architecture on data derived from the Metamath library (Megill & Wheeler, 2019) for proof synthesis. PACT expands on GPT-$f$ by incorporating auxiliary training tasks for the neural module towards theorem-proving in Lean 3. FMSCL (Polu et al., 2022) alternates proof-search and training to finetune their neural model based on proofs found during search. HTPS (Lample et al., 2022) uses a transformer-based neural module in an online, MCTS-inspired proof search in Lean 3 and Metamath. COPRA (Thakur et al., 2024) uses GPT-4 supplied with error feedback from the environment and lemmas from a retrieval mechanism for an agentic proof-search in Lean 3 and Coq. LLEMMA (Azerbayev et al., 2024) continues pretraining of Code Llama on a mathematics-based corpus dubbed Proof-Pile-2, and uses their learned model for formal proof search in Lean 4. DeepSeek-Prover Xin et al. (2024) produces synthetic Lean data en-masse for training their prover model. AlphaGeometry (Trinh et al., 2024) targets IMO problems in a geometry-specific proof assistant language using an interleaving search, where a neural module synthesizes auxiliary constructions and a symbolic engine produces deductive closures.

The Isabelle proof assistant (Paulson, 1994), given its declarative nature and powerful symbolic automation, has too been the focus of much attention for neural theorem proving. Isabelle features Sledgehammer (Paulson & Blanchette, 2015), an automated reasoning tool which calls external automated theorem provers (ATPs) for proof synthesis. Draft, Sketch, Prove (DSP) (Jiang et al., 2022b) uses a high-caliber LLM to generate natural language proofs and converts them into formal *sketches* in Isabelle, whose gaps are then filled using Sledgehammer. Zhao et al. (2023) employed a diffusion model to predict an optimal ordering of the few-shot examples provided to the LLM in the DSP pipeline. Lyra (Zheng et al., 2023) utilized error-feedback from Isabelle's execution to modify holes in the sketch which were too difficult for the symbolic prover. POETRY (Wang et al., 2024) leverages recursion for theorem-proving and trains a neural module to produce proof sketches, as opposed to using in-context learning with an LLM. LEGO-Prover (Wang et al., 2023) extends the pipeline by incorporating a skill library which grows throughout the proof search task. Separate from approaches utilizing natural language proofs, Thor (Jiang et al., 2022a) trains a transformer-based architecture to predict successful invocations of Sledgehammer, along with the usual proof-step objective. Baldur (First et al., 2023) explored repairing erroneous proofs in Isabelle through the use of LLMs.

The Coq interactive theorem prover has seen use in both software verification and general mathematics. Famously, mechanized proofs of the Four Colour Theorem (Robertson et al., 1997) and the Feit-Thompson theorem (Gonthier et al., 2013) were produced in Coq. Similarly, numerous software verification projects have been undertaken in Coq, such as CompCert (a formally verified C compiler) and Verdi (Wilcox et al., 2015) (a framework for verifying distributed systems protocols). ASTactic (Yang & Deng, 2019) trained a neural module involving recurrent networks and attention on data collected from various Coq repositories. Proverbot9001 (Sanchez-Stern et al., 2020) targeted proof synthesis on a set of held-out theorems from the CompCert project. COPRA (Thakur et al., 2024) also evaluates on this CompCert-based task using their multi-language approach. Tactician (Blaauwbroek et al., 2020) develops a platform for proof automation for the Coq practitioner, with support for experimenting with new machine learning techniques for tactic prediction and proof search. (Zhang et al., 2021) explores several online learning techniques inside Tactician, including an approximate $k$-nearest neighbors method via locality sensitive hashing which we use for our evaluation. Graph2Tac (Blaauwbroek et al., 2024) uses graph neural networks for learning online hierarchical representations of new theorems and definitions, and is used for proof search within Tactician.

# 6 Conclusion

We presented PUTNAMBENCH, a benchmark for neural theorem-proving consisting of formalizations of Putnam competition problems. A distinctive feature of PUTNAMBENCH is that it spans a broad range of undergraduate-level mathematical topics, including algebra, analysis, and number theory. Another unique benefit is that it includes problems in Lean 4, Isabelle, and Coq, the three most popular formal proof frameworks.

As our experiments show, PUTNAMBENCH is a challenging benchmark: all current theorem-proving approaches fail to solve more than a handful of its problems. We believe that these failures include two root causes: (i) While current theorem-provers can effectively stitch together standard proof steps well-represented in the training corpus, they often fail at synthesizing new lemmas and orchestrating these lemmas into intricate proofs. (ii) Current methods often fail to leverage the deep knowledge available in mathematics repositories. Developing a new generation of neural theorem-provers in which these weaknesses are at least partly addressed is an exciting direction of future research.

**Acknowledgements.** This work was supported by NSF awards CCF-2212559 and CCF-2403211, the NSF Institute for Foundations of Machine Learning, and a gift by the Aziz Family Foundation. We thank Oliver Nash, Eric Wieser, Edward Lockhart, Fabian Gloeckle, Karl Palmskog, Lasse Blaauwbroek, Jason Rute, and Kaiyu Yang for useful discussions, aiding in benchmark maintenance, and support with setting up experiments.

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

# A Appendix

We include further examples of formalizations from PUTNAMBENCH below.

```
From mathcomp Require Import ssrbool seq ssrnat prime rat ssralg ssrnum ssrint.

Local Open Scope ring_scope.

Theorem putnam_2009_b1 :
  let fact_prod (ls : seq nat) : rat := \prod_(i <- ls) (i`!)%:Q in
  forall q : rat, q > 0 -> exists n d : seq nat,
  all prime (n ++ d) /\ fact_prod n / fact_prod d = q.
Proof. Admitted.
```

Figure 5: A formalization of Putnam 2009 B1 in Coq relying on the MathComp repository.

**Putnam 2001 B4.** Let $S$ denote the set of rational numbers different from $\{-1, 0, 1\}$. Define $f : S \to S$ by $f(x) = x - 1/x$. Prove or disprove that

$$\bigcap_{n=1}^{\infty} f^{(n)}(S) = \varnothing,$$

where $f^{(n)}$ denotes $f$ composed with itself $n$ times.

```
abbrev putnam_2001_b4_solution : Prop := True
theorem putnam_2001_b4
    (S : Set ℚ)
    (hS : S = univ \ {-1, 0, 1})
    (f : S → S)
    (hf : ∀ x : S, f x = x - 1 / (x : ℚ))
    : ∩ n ∈ Ici 1, f^[n] '' univ = ∅ ↔ putnam_2001_b4_solution
    := sorry
```

Figure 6: A formalization of Putnam 2001 B4 in Lean 4. As the problem requires deciding whether the infinite intersection is empty, it is not directly the statement of a theorem. We consider the associated "solution" of this problem to be a boolean value, and factor it out from the theorem statement. sorry is the placeholder keyword for Lean.

**Putnam 2020 A3.** Let $a_0 = \pi/2$, and let $a_n = \sin(a_{n-1})$ for $n \geq 1$. Determine whether

$$\sum_{n=1}^{\infty} a_n^2$$

converges.

```
abbrev putnam_2020_a3_solution : Prop := False
theorem putnam_2020_a3
    (a : ℕ → ℝ)
    (ha0 : a 0 = Real.pi / 2)
    (ha : ∀ n : ℕ, n ≥ 1 → a n = Real.sin (a (n - 1)))
    : (∃ L : ℝ, Tendsto (fun m : ℕ => Σ n : Icc 1 m, (a n)^2) atTop (N L))
        ↔ putnam_2020_a3_solution
    := sorry
```

Figure 7: A formalization of Putnam 2020 A3 in Lean 4. As the problem requires deciding whether the series converges, it is not directly the statement of a theorem. We consider the associated "solution" of this problem to be a boolean value, and factor it out from the theorem statement.

**Putnam 1997 A4.** Let $G$ be a group with identity $e$ and $\phi : G \to G$ a function such that

$$\phi(g_1)\phi(g_2)\phi(g_3) = \phi(h_1)\phi(h_2)\phi(h_3)$$

whenever $g_1 g_2 g_3 = e = h_1 h_2 h_3$. Prove that there exists an element $a \in G$ such that $\psi(x) = a\phi(x)$ is a homomorphism.

```
theorem putnam_1997_a4
    (G : Type*)
    [Group G]
    (φ : G → G)
    (hφ : ∀ g1 g2 g3 h1 h2 h3 : G, (g1 * g2 * g3 = 1 ∧ h1 * h2 * h3 = 1)
    → φ g1 * φ g2 * φ g3 = φ h1 * φ h2 * φ h3)
    : ∃ a : G, let ψ := fun g => a * φ g; ∀ x y : G, ψ (x * y) = ψ x * ψ y
    := sorry
```

Figure 8: A formalization of Putnam 1997 A4, which requires knowledge of group theory, in Lean 4. The informal statement is slightly underspecified - $g_1, g_2, g_3, h_1, h_2, h_3$ are not explicitly defined to be in $G$. To produce the formalization, we must be specific about the type of $g_i, h_i$.

**Putnam 2018 B1.** Let $\mathcal{P}$ be the set of vectors defined by

$$\mathcal{P} = \left\{ \begin{pmatrix} a \\ b \end{pmatrix} \middle| \, 0 \leq a \leq 2, 0 \leq b \leq 100, \text{ and } a, b \in \mathbb{Z} \right\}$$

Find all $\mathbf{v} \in \mathcal{P}$ such that the set $\mathcal{P} \setminus \{\mathbf{v}\}$ obtained by omitting vector $\mathbf{v}$ from $\mathcal{P}$ can be partitioned into two sets of equal size and equal sum.

```
abbrev putnam_2018_b1_solution : Set (Vector ℤ 2) :=
    {v : Vector ℤ 2 | ∃ b : ℤ, 0 ≤ b ∧ b ≤ 100 ∧ Even b ∧ v.toList = [1, b]}
theorem putnam_2018_b1
(v : Mathlib.Vector ℤ 2)
(P Pvdiff : Finset (Mathlib.Vector ℤ 2))
(hP : P =
    {v' : Mathlib.Vector ℤ 2 | 0 ≤ v'[0] ∧ v'[0] ≤ 2 ∧ 0 ≤ v'[1] ∧ v'[1] ≤ 100})

(hPvdiff : Pvdiff = P \ ({v} : Finset (Mathlib.Vector ℤ 2)))
: (v ∈ P ∧ (∃ Q R : Finset (Mathlib.Vector ℤ 2),
    (Q ∪ R = Pvdiff) ∧ (Q ∩ R = ∅) ∧ (Q.card = R.card) ∧
    (Σ q in Q, q[0] = Σ r in R, r[0]) ∧ (Σ q in Q, q[1] = Σ r in R, r[1])))
  ↔ v ∈ putnam_2018_b1_solution :=
sorry
```

Figure 9: A formalization of Putnam 2018 B1, which requires the Vector class from mathlib4.

**Putnam 1992 B6.** Let $\mathcal{M}$ be a set of real $n \times n$ matrices such that

1. $I \in \mathcal{M}$, where $I$ is the $n \times n$ identity matrix;
2. if $A \in \mathcal{M}$ and $B \in \mathcal{M}$, then exactly one of $AB \in \mathcal{M}$ and $-AB \in \mathcal{M}$ holds;
3. if $A \in \mathcal{M}$ and $B \in \mathcal{M}$, then either $AB = BA$ or $AB = -BA$;
4. if $A \in \mathcal{M}$ and $A \neq I$, there is at least one $B \in \mathcal{M}$ such that $AB = -BA$.

Prove that $\mathcal{M}$ contains at most $n^2$ matrices.

```
theorem putnam_1992_b6:
  fixes n :: nat
    and M :: "(real^'n^'n) set"
  assumes npos: "n > 0"
    and pncard: "CARD('n) = n"
    and h1: "mat 1 ∈ M"
    and h2: "∀A∈M. ∀B∈M. (A**B ∈ M) ≠ (-A**B ∈ M)"
    and h3: "∀A∈M. ∀B∈M. (A**B = B**A) ∨ (A**B = -B**A)"
    and h4: "∀A∈M. (A ≠ mat 1 → (∃B∈M. A**B = -B**A))"
  shows "card M ≤ n^2"
  sorry
```

Figure 10: An Isabelle formalization of Putnam 1992 B6.

**Putnam 2012 A3.** Let $f : [-1, 1] \to \mathbb{R}$ be a continuous function such that

1. $f(x) = \frac{2-x^2}{2} f(\frac{x^2}{2-x^2})$ for every $x$ in $[-1, 1]$,

2. $f(0) = 1$, and

3. $\lim_{x \to 1^-} \frac{f(x)}{\sqrt{1-x}}$ exists and is finite.

Prove that $f$ is unique, and express $f(x)$ in closed form.

```
definition putnam_2012_a3_solution :: "real ⇒ real" where
  "putnam_2012_a3_solution ≡ (λx::real. sqrt (1 - x^2))"
theorem putnam_2012_a3:
  fixes S :: "real set"
  and hf :: "(real ⇒ real) ⇒ bool"
  defines "S ≡ {-1..1}"
  and "hf ≡ (λf::real⇒real. continuous_on S f ∧
    (∀x∈S. f x = ((2 - x^2)/2)*f (x^2/(2 - x^2))) ∧ f 0 = 1 ∧
    (∃y::real. filterlim (λx::real. (f x)/sqrt (1 - x)) (nhds y) (at_left 1)))"
  shows "hf putnam_2012_a3_solution ∧
    (∀f::real⇒real. hf f → (∀x∈S. f x = putnam_2012_a3_solution x))"
  sorry
```

Figure 11: An Isabelle formalization of Putnam 2012 A3. The mechanism for factoring the solution out of the theorem statement is similar to that of Lean.

**Putnam 1980 A5.** Let $P(t)$ be a nonconstant polynomial with real coefficients. Prove that the system of simultaneous equations

$$0 = \int_0^x P(t) \sin t\, dt = \int_0^x P(t) \cos t\, dt$$

has only finitely many real solutions $x$.

```
From mathcomp Require Import all_algebra all_ssreflect.
From mathcomp Require Import reals trigo lebesgue_integral lebesgue_measure measure.

From mathcomp Require Import classical_sets cardinality.

Set Implicit Arguments.
Unset Strict Implicit.
Unset Printing Implicit Defensive.

Local Open Scope classical_set_scope.
Local Open Scope ring_scope.

Variable R : realType.
Definition mu := [the measure _ _ of @lebesgue_measure R].
Theorem putnam_1980_a5
    (P : {poly R})
    (Pnonconst : gtn (size P) (1%nat))
    : finite_set [set x : R |
    \int[mu]_(t in [set t : R | 0 <= t <= x]) (fun y => P.[y] * (sin y)) t = 0 /\
    \int[mu]_(t in [set t : R | 0 <= t <= x]) (fun y => P.[y] * (cos y)) t = 0].
Proof. Admitted.
```

Figure 12: A Coq formalization of Putnam 1980 A5. This formalization is done using Coquelicot, a Coq repository outside of the standard library. The Coq equivalent of sorry is Admitted.

**Putnam 2017 B2.** Suppose that a positive integer $N$ can be expressed as the sum of $k$ consecutive positive integers
$$N = a + (a+1) + (a+2) + \cdots + (a+k-1)$$
for $k = 2017$ but for no other values of $k > 1$. Considering all positive integers $N$ with this property, what is the smallest positive integer $a$ that occurs in any of these expressions?

```
From mathcomp Require Import all_ssreflect all_algebra.

Set Implicit Arguments.
Unset Strict Implicit.
Unset Printing Implicit Defensive.

Local Open Scope ring_scope.

Definition putnam_2017_b2_solution : nat := 16.
Theorem putnam_2017_b2 :
    let seq (a : int) (k : nat) := \sum_(0 <= i < k) (a + i%:Z) in
    let valid (a : int) := a > 0 /\ (forall (b : int) (k : nat), b > 0 -> gt k 1 ->
      seq a 2017%nat = seq b k -> k = 2017%nat) in
    valid putnam_2017_b2_solution /\ (forall a, valid a -> a >=
    putnam_2017_b2_solution%:Z).
Proof. Admitted.
```

Figure 13: A Coq formalization of Putnam 2017 B2. As the problem requires a numerical witness, we factor that out using Coq's syntax for making definitions.

**Putnam 1988 B1.** A *composite* is a product $ab$ with $a$ and $b$ not necessarily distinct integers $\{2, 3, 4, \dots\}$. Show that every composite is expressible as $xy + xz + yz + 1$ with $x, y, z$ positive integers.

```
Require Import ZArith Znumtheory Lia.
Open Scope Z.
Theorem putnam_1988_b1:
    forall (a : Z), a >= 2 ->
    forall (b : Z), b >= 2 ->
    exists (x y z: Z), x > 0 /\ y > 0 /\ z > 0 /\
    a * b = x * y + y * z + z * x + 1.
Proof.
    intros a Ha b Hb.
    exists 1, (a - 1), (b - 1).
    split.
    - lia.
    - split.
    + lia.
    + split.
    * lia.
Qed.
```

Figure 14: A Coq proof of Putnam 1988 B1 generated through a few-shot invocation of GPT-4. The proof is similar to that of the Lean version, also discovered by GPT-4. The main difficulty of the problem is to choose the values of $x, y, z$ given $a, b$. Once correctly supplied, the remainder of the proof is routine and can be done with automated methods like lia which handles linear arithmetic.

```
theorem mathd_numbertheory_85 :
  1 * 3^3 + 2 * 3^2 + 2*3 + 2 = 53
  := sorry
```

```
theorem mathd_algebra_107
(x y : ℝ)
(h₀ : x^2 + 8 * x + y^2 - 6 * y = 0)
: (x + 4)^2 + (y-3)^2 = 5^2 := sorry
```

Figure 15: Examples of formalizations of easy problems in MINIF2F. While useful for benchmarking straightforward mathematical reasoning in a formal setting, these problems are quite simple compared to the competition problems present in PUTNAMBENCH. We note that MINIF2F does include some formalizations of problems sourced directly from high school competitions, but these are fewer in number.

```
You are proficient at formal theorem-proving in Lean 4. Given a theorem
↪  statement in Lean 4, generate the proof in Lean 4. You can assume that
↪  you have access to Lean's mathlib library.

The theorem is described in the following format:
1. The theorem statement using the `[THEOREM]` keyword.
3. The theorem description ends with the keyword `[END]`.

Generate a Lean 4 proof for the theorem which starts with the keyword
↪  `[PROOF]` followed by the proof of the theorem. The syntax for Lean 4
↪  is different than that of Lean 3 - premises like "Nat.dvd_mul" and
↪  "Finset.singleton_injective" exist in Lean 4, the equivalent in Lean 3
↪  is "nat.dvd_mul" and "finset.singleton_injective" which DO NOT WORK in
↪  Lean 4. Additionally, you cannot chain tactics into one step using ',' -
↪  this will NOT work - you can use ';' instead but try to avoid such usage
↪  where not necessary! When doing rewrites you MUST wrap the premise in
↪  brackets: "rw [h]". If you want to do multiple rewrites at once you can
↪  do something like "rw [step1, step2, step3]". Always predict one tactic
↪  at a time, though you can predict the "have" tactic and may supply a
↪  proof for it with tactics split by ";". You can provide witnesses to
↪  consecutive existential quantifiers all at once, for example 'use 1, 2,
↪  3' but NOT as a list 'use [1, 2, 3]' - these are not the same things!
↪  You can introduce with "intro" everything you think you can introduce at
↪  once. In Lean 4, you can split apart conjunctions with "constructor" NOT
↪  "split". You should use the "ring" tactic to handle goals that follow
↪  from ring axioms, especially instead of doing a long series of rewrites
↪  or calculations. Similarly, "linarith" can be useful for solving goals
↪  involving linear arithmetic. Do NOT indent tactics, every new line
↪  should not have spaces to start! PLEASE use Lean 4 syntax only! The
↪  proof ends with the keyword `[END]`. Also please DO NOT write `sorry`
↪  in the proof. You can assume that the theorem is provable.
```

Figure 16: Parts of the "system prompt" used by GPT-4 for Lean 4 evaluations. Due to GPT-4's tendency towards producing outputs in Lean 3 syntax, our prompt places special attention towards preventing such syntactic mistakes. A similar modification is made to COPRA's system prompt for Lean 3.

```
Goals to prove:
[GOALS]
[GOAL] 1
DifferentiableAt ℝ (fun x => g x / hg0 x) 0 → DifferentiableAt ℝ g 0
[HYPOTHESES] 1
[HYPOTHESIS] case mpr
[HYPOTHESIS] f : True
[HYPOTHESIS] g hg0 : ℝ → ℝ
[HYPOTHESIS] hcg : hg0 0 ≠ 0
[HYPOTHESIS] hfg : ContinuousAt hg0 0
[HYPOTHESIS] hfg_div : DifferentiableAt ℝ (fun x => g x * hg0 x) 0

[STEPS]
[STEP] constructor
[STEP] intro h
[STEP] trivial
[STEP] intros f g hg0 hcg hfg hfg_div

[INCORRECT STEPS]
[STEP] apply differentiable_at_of_mul

[LAST STEP]
apply differentiable_at.div

[ERROR MESSAGE]
error: unknown identifier 'differentiable_at.div'
[END]
```

Figure 17: An example of a failed tactic prediction during proof search for Putnam 2011 B2 using COPRA in Lean 4. GPT-4 predicts a tactic involving the premise "differentiable_at.div," which exists in Lean 3, but not Lean 4. Even with the system prompt asserting outputs should involve Lean 4 syntax alone, GPT-4 is not always capable of making the distinction.

```
theorem putnam_2001_a1
(S : Type*)
[Mul S]
(hS : ∀ a b : S, (a * b) * a = b)
: ∀ a b : S, a * (b * a) = b := by
intro a b
have h1 : (a * (b * a)) * a = b * a := by
 rw ←[mul_assoc]
 rw [hS]
 rw [hS]
 apply h1
```

Figure 18: A failed proof generated by few-shot invocation of GPT-4. GPT-4 misunderstands that the hypothesis [Mul S], which gives an operation ⋆ and asserts it is a binary operation on $S$, also asserts associativity of the operation. The tactic rw [←mul_assoc], which performs a rewrite using the associativity of ⋆, is hence not applicable in this setting.

**Putnam 1986 B1.** Inscribe a rectangle of base $b$ and height $h$ in a circle of radius one, and inscribe an isosceles triangle in the region of the circle cut off by one base of the rectangle. For what values of $h$ do the rectangle and triangle have the same area?

```
theorem putnam_1986_b1:
fixes b h :: "real"
assumes hbh : "b > 0 ∧ h > 0 ∧ b^2 + h^2 = 2^2"
and areaeq : "b*h = 0.5 * b * (1 - h/2)"
shows "h = ( 2 / 5 )"
proof -
from assms have eq1: "b^2 + h^2 = 4" by simp
  from areaeq have eq2: "b * h = 0.5 * b * (1 - h/2)" by simp
  then have eq3: "b * h = 0.5 * b - 0.25 * b * h" by algebra
  then have eq4: "1.25 * b * h = 0.5 * b" by algebra
  then have eq5: "h = 0.5 / 1.25" using assms(1) by simp
  then show ?thesis by simp
qed
```

Figure 19: A successful proof in Isabelle discovered using GPT-4. While the theorem statement differs from the formalization, which is algebraic, we note that the official solution is also algebraic and is similar to the generated Isabelle proof.

**Putnam 1995 A1.** Let $S$ be a set of real numbers which is closed under multiplication (that is, if $a$ and $b$ are in $S$, then so is $ab$. Let $T$ and $U$ be disjoint subsets of $S$ whose union is $S$. Given that the product of any *three* (not necessarily distinct) elements of $T$ is in $T$ and the product of any three elements of $U$ is in $U$, show that at least one of the two subsets $T, U$ is closed under multiplication.

```
theorem putnam_1995_a1:
  fixes S :: "real set"
    and T U :: "real set"
  assumes hS: "∀a∈S. ∀b∈S. a * b ∈ S"
    and hsub: "T ⊆ S ∧ U ⊆ S"
    and hunion: "(T ∪ U) = S"
    and hdisj: "(T ∩ U) = {}"
    and hT3: "∀a∈T. ∀b∈T. ∀c∈T. a * b * c ∈ T"
    and hU3: "∀a∈U. ∀b∈U. ∀c∈U. a * b * c ∈ U"
  shows "(∀a∈T. ∀b∈T. a * b ∈ T) ∨ (∀a∈U. ∀b∈U. a * b ∈ U)"
proof (rule ccontr)
  assume "¬ ((∀a∈T. ∀b∈T. a * b ∈ T) ∨ (∀a∈U. ∀b∈U. a * b ∈ U))"
  then have "∃t1∈T. ∃t2∈T. t1 * t2 ∉ T" "∃u1∈U. ∃u2∈U. u1 * u2 ∉ U" by auto
  then obtain t1 t2 where t1t2: "t1 ∈ T" "t2 ∈ T" "t1 * t2 ∈ U"
    using hS hunion by blast
  then obtain u1 u2 where u1u2: "u1 ∈ U" "u2 ∈ U" "u1 * u2 ∈ T"
    using ‹¬ ((∀a∈T. ∀b∈T. a * b ∈ T) ∨ (∀a∈U. ∀b∈U. a * b ∈ U))› hS hunion
    by blast
  have "t1 * t2 * t3 ∈ T" if "t3 ∈ T" for t3 using hT3 t1t2(1) t1t2(2) that
    by auto
  then have "t1 * t2 * t3 ∈ U" if "t3 ∈ T" for t3 using t1t2(3) hS that
    by (metis disjoint_iff hU3 hdisj mult.commute u1u2(1) u1u2(2) u1u2(3))
  then have False using hdisj t1t2(1) t1t2(2)
    by (meson disjoint_iff_not_equal hT3)
  then show False by blast
qed
```

Figure 20: A successful proof in Isabelle discovered using DSP. In particular, this proof is not found using a single invocation of sledgehammer, so the sketching mechanism of DSP is crucial for this problem. We note that the DSP pipeline involves using an LLM (GPT-4) to synthesize an informal proof which is translated into a sketch in Isabelle - this can potentially be a source of indirect dataset contamination, as we cannot ensure the informal proofs are not present in GPT-4's training data.

```
theorem putnam_1971_b1:
  fixes Smul :: "'S ⇒ 'S ⇒ 'S" (infixl "*" 70)
  assumes hself: "∀x::'S. x * x = x"
    and h2: "∀x y z::'S. (x * y) * z = (y * z) * x"
  shows "∀x y z::'S. (x * y) *  z = x * (y * z) ∧ x * y = y * x"
proof -
  have comm: "∀x y::'S. x * y = y * x"
  proof
    fix x y :: 'S
    have "(x * y) * x = (y * x) * x" using h2 by blast (* sledgehammer *)
    also have "... = y * x" using hself by (metis h2) (* sledgehammer *)
    finally have "(x * y) * x = y * x" by simp (* sledgehammer *)
    then have "x * y = y * x" using hself by (metis h2) (* sledgehammer *)
    thus "x * y = y * x" by simp
  qed
  have assoc: "∀x y z::'S. (x * y) * z = x * (y * z)"
  proof
    fix x y z :: 'S
    have "(x * y) * z = (y * z) * x" using h2 sledgehammer
    also have "... = x * (y * z)" using comm sledgehammer
    finally show "(x * y) * z = x * (y * z)" by simp
  qed
  show "∀x y z::'S. (x * y) * z = x * (y * z) ∧ x * y = y * x"
    using assoc comm by simp
qed
```

Figure 21: An erroneous generation produced by DSP for Putnam 1971 B1. While a single invocation of Sledgehammer can prove this problem, the formal sketch generated in the pipeline is much more complex and is erroneous, leading to a failed proof attempt.

```
theorem putnam_1971_b1:
  fixes Smul :: "'S ⇒ 'S ⇒ 'S" (infixl "*" 70)
  assumes hself: "∀x::'S. x * x = x"
  and h2: "∀x y z::'S. (x * y) * z = (y * z) * x"
  shows "∀x y z::'S. (x * y) * z = x * (y * z) ∧ x * y = y * x"
```

```
theorem putnam_2012_a2:
  fixes Smul :: "'S ⇒'S ⇒ 'S" (infixl "*" 70)
  and a b c :: 'S
  assumes Smulasg: "abel_semigroup Smul"
  and hS: "∀x y::'S. ∃z::'S. x * z = y"
  and habc: "a * c = b * c"
shows "a = b"
```

Figure 22: The other two Isabelle formalizations solved with invocations of Sledgehammer. We note that the problems Sledgehammer was capable of solving are all problems involving binary operations on sets. It is not surprising that SMT solvers are capable of solving such problems, which do not require reasoning about complicated objects.

```
Goals to prove:
[GOALS]
[GOAL] 1
1 > 0 /\
1 > 0 /\
a * b - 1 > 0 /\ a * b = 1 * 1 + 1 * (a * b - 1) + (a * b - 1) * 1 + 1
[HYPOTHESES] 1
[HYPOTHESIS] a : Z
[HYPOTHESIS] Ha : a >= 2
[HYPOTHESIS] b : Z
[HYPOTHESIS] Hb : b >= 2

[STEPS]
[STEP] intros a Ha b Hb.

[LAST STEP]
exists 1, 1, (a * b - 1).

[SUCCESS]
[END]
...
Goals to prove: # Step 32 of Search
[GOALS]
[GOAL] 1
a - 1 > 0 /\
1 > 0 /\
b - 1 > 0 /\ a * b = (a - 1) * 1 + 1 * (b - 1) + (b - 1) * (a - 1) + 1
[HYPOTHESES] 1
[HYPOTHESIS] a : Z
[HYPOTHESIS] Ha : a >= 2
[HYPOTHESIS] b : Z
[HYPOTHESIS] Hb : b >= 2

[STEPS]
[STEP] intros a Ha b Hb.

[LAST STEP]
exists (a - 1), 1, (b - 1).

[SUCCESS]
[END]
```

Figure 23: Early in COPRA's attempt on Putnam 1988 B1, an incorrect prediction of $x, y, z$ given $a, b$ is made, which dooms that path of search as the most crucial step is the correct choice. Later, at step 32 of search, COPRA backtracks and then successfully predicts a correct choice for $x, y, z$. Once this step is generated, the remainder of the proof is straightforward.

