# OpenReview forum: "PutnamBench: Evaluating Neural Theorem-Provers on the Putnam Mathematical Competition"
_NeurIPS.cc/2024/Datasets_and_Benchmarks_Track — NeurIPS 2024 Track Datasets and Benchmarks Poster_

### Official Review · Reviewer_pTSB · 2024-06-20
**The proposed PutnamBench is very exciting and has the potential to serve as a fundamental benchmark for evaluating neural theorem provers.**

**Rating:** 8
**Confidence:** 4
**Correctness:** Most claims made in this paper are co…
**Clarity:** The paper is generally well-written a…

**Review:**

This paper provides a multi-language, multi-domain formal theorem-proving benchmark, PutnamBench, with 514 problems drawn from undergraduate-level math competitions. The evaluation shows that PutnamBench is indeed very challenging, as existing neural and symbolic methods can only solve 5 problems in total. I greatly appreciate the manual efforts involved in constructing this dataset and believe it could significantly benefit future research in this field. For a detailed list of pros and cons, please see the sections below.

* Quality: 4/5

* Clarity: 4/5

* Originality: 4/5

* Significance: 5/5

**Strengths:**

* The paper is well-motivated and well-written.

* The proposed PutnamBench is the first challenging benchmark targeting the formalization of undergraduate-level math competitions, covering a wide array of math categories. It provides problems in three different formal languages, and its size is larger than the widely used miniF2F benchmark, which targets high-school math competitions. The manual efforts are commendable, and I believe PutnamBench could serve as a fundamental benchmark for future studies.

* The evaluation and discussion of three neural baselines and hammer tools on PutnamBench are detailed and convincing.

**Additional Feedback:**

I also have a few questions and would be happy to discuss them with the authors:

* Why does DSP underperform compared to Sledgehammer? DSP also calls Sledgehammer to fill the proof gaps in the formal sketch. Could you provide some error analysis of the cases where DSP fails to prove but Sledgehammer succeeds?

* miniF2F also provides some formal proofs if applicable. I am curious about the difficulty of manually proving the instances in PutnamBench. Could the easiest instance be proved manually in just a few steps?

* When using DSP or COPRA, one first needs to produce an informal proof. I am also curious about the informal proof accuracy by GPT-4. Can it correctly prove the easiest problem in PutnamBench?

**Documentation:**

The proposed dataset is well-documented.

**Ethics:**

There are no ethical concerns.

**Limitations:**

The authors mention that one limitation of PutnamBench is the challenge of formalizing certain problem categories such as geometry and probability. Moreover, I think the evaluation setting could be more detailed. For example, it seems unclear whether task 1 or task 2 is used in the evaluation.

**Opportunities For Improvement:**

I would say the paper is overall very good, but there are also a few potential improvements that could be considered:

* The term 'multilingual' may not be accurate. 'Multi-language' may be better in this case to refer to more than one formal language.

* The discussion of miniF2F may not be entirely accurate. miniF2F indeed targets some other languages (Metamath, HOL Light) besides Lean and Isabelle.

* Given that PutnamBench also provides natural language statements, it would be interesting to evaluate the autoformalization capabilities of LLMs on PutnamBench.

* To mimic human performance in the Putnam Competition, it would also be interesting to set a time (instead of #attempts) limit for these neural baselines similar to that given to students, which aligns with the AIMO and IMO Grand Challenge.

**Relation To Prior Work:**

The authors discussed how this work differs from previous contributions.

**Summary And Contributions:**

This paper proposes PutnamBench, a multi-language and challenging theorem-proving benchmark targeting undergraduate-level math competitions. PutnamBench draws 514 problems from the Putnam Competition, covering a wide range of domains. The authors manually formalize the natural language problems into both Lean and Isabelle, and a large portion into Coq, resulting in 1,337 formalizations in total. PutnamBench also factors out solutions and introduces a new task: requiring the theorem prover first to identify the (closed-form) solution and then provide a proof of correctness by rewriting the solution into the theorem statement. Experiments on three neural baselines (GPT-4, DSP, COPRA) and two hammer systems (Sledgehammer, CoqHammer) demonstrate that PutnamBench is very challenging, with only 5 problems being solved by these methods.

---

> ### Author Rebuttal · Authors · 2024-08-15
>
> Thank you for your feedback and suggestions. We appreciate the kind words regarding our effort! We address the main suggestions and questions below. We are happy to discuss more, please let us know if you have further questions or comments!
>
> >  The term 'multilingual' may not be accurate. 'Multi-language' may be better in this case to refer to more than one formal language.
>
> We agree, ‘multilingual’ can be confusing as it might be interpreted as referring to several spoken languages, not formal languages. We can modify this in a subsequent revision.
>
> > The discussion of miniF2F may not be entirely accurate. miniF2F indeed targets some other languages (Metamath, HOL Light) besides Lean and Isabelle.
>
> We did mention in the related works lines 312-314 that miniF2F targets Metamath & HOL Light and our reason for not incorporating those languages in our benchmark. There has not been a significant interest in the Metamath & HOL Light formalizations from miniF2F in the recent AI-for-math literature. We wanted to choose the languages which are of most interest amongst the research community. Predominantly, those have been Lean & Isabelle. Coq also has a large user community, though has not yet had a benchmark for competition mathematics.
>
> > Given that PutnamBench also provides natural language statements, it would be interesting to evaluate the autoformalization capabilities of LLMs on PutnamBench.
>
> We agree! This can be a particularly enlightening evaluation space, especially considering that formalizing some Putnam competition problems requires heavy semantic parsing. Since there can be many formalizations of a given problem, exact-match accuracy is not a reliable indicator of autoformalization ability. Hence evaluating autoformalization on these problems with LLMs must be done carefully with manual annotators. We think the autoformalization task in the case of PutnamBench is an interesting avenue for future work.
>
> > To mimic human performance in the Putnam Competition, it would also be interesting to set a time (instead of #attempts) limit for these neural baselines similar to that given to students, which aligns with the AIMO and IMO Grand Challenge.
>
> We agree! In the case of our baselines in the paper, producing evaluations that run within say, 30 minutes, can become prohibitive cost-wise and time-wise, especially considering the number of problems in the benchmark. Sampling GPT4 as many times as possible within such a time frame will quickly exceed our budget.
>
> We think your point will become more important as more methods appear which can solve more than a handful of problems. A high attempt count can be seen as a weak approximation of time spent - on the leaderboard we're hosting, a team submitted their result using pass@4096: https://trishullab.github.io/PutnamBench/leaderboard.html. We think it is possible to have a variant of the IMO Grand Challenge using the next set of 12 problems from the upcoming Putnam competition in December (which we plan to incorporate in the benchmark), and are planning to work to host such a challenge in the future.
>
> > Why does DSP underperform compared to Sledgehammer? DSP also calls Sledgehammer to fill the proof gaps in the formal sketch. Could you provide some error analysis of the cases where DSP fails to prove but Sledgehammer succeeds?
>
> First, we would like to mention that since the submission, we have incorporated ~100 more formalizations in each language. PutnamBench now consists of all problems we can reasonably formalize from years 1962-2023. With this modification, we have also reran the baselines on all new problems and those which were previously solved. The new results (for Isabelle) are:
>
> GPT-4 (1/640); DSP (4/640); Sledgehammer (3/640)
>
> Sledgehammer is capable of solving all three problems involving simple reasoning about magmas. DSP can solve an additional problem which requires the high-level sketch with steps that can be filled in by Sledgehammer. We include this below:
>
> ```
> theorem putnam_1995_a1:
> fixes S :: "real set"
> and T U :: "real set"
> assumes hS: "∀a∈S. ∀b∈S. a * b ∈ S"
> and hsub: "T ⊆ S ∧ U ⊆ S"
> and hunion: "(T ∪ U) = S"
> and hdisj: "(T ∩ U) = {}"
> and hT3: "∀a∈T. ∀b∈T. ∀c∈T. a * b * c ∈ T"
> and hU3: "∀a∈U. ∀b∈U. ∀c∈U. a * b * c ∈ U"
> shows "(∀a∈T. ∀b∈T. a * b ∈ T) ∨ (∀a∈U. ∀b∈U. a * b ∈ U)"
>
> proof (rule ccontr)
> assume "¬ ((∀a∈T. ∀b∈T. a * b ∈ T) ∨ (∀a∈U. ∀b∈U. a * b ∈ U))"
> then have "∃t1∈T. ∃t2∈T. t1 * t2 ∈/ T" "∃u1∈U. ∃u2∈U. u1 * u2 ∈/ U" by (sledgehammer)
> then obtain t1 t2 where t1t2: "t1 ∈ T" "t2 ∈ T" "t1 * t2 ∈ U"
> using hS hunion by (sledgehammer)
> then obtain u1 u2 where u1u2: "u1 ∈ U" "u2 ∈ U" "u1 * u2 ∈ T"
> using ‹¬ ((∀a∈T. ∀b∈T. a * b ∈ T) ∨ (∀a∈U. ∀b∈U. a * b ∈ U))› hS hunion
> by (sledgehammer)
> have "t1 * t2 * t3 ∈ T" if "t3 ∈ T" for t3 using hT3 t1t2(1) t1t2(2) that
> by (sledgehammer)
> then have "t1 * t2 * t3 ∈ U" if "t3 ∈ T" for t3 using t1t2(3) hS that
> by (sledgehammer)
> then have False using hdisj t1t2(1) t1t2(2)
> by (sledgehammer)
> then show False by (sledgehammer)
> qed
> ```
>
> In general, Sledgehammer is not directly subsumed by DSP. If Sledgehammer is called to fill in intermediate goals in a sketch, rather than the final goal (which is the problem statement), the proof attempt fails if Sledgehammer fails for any of the intermediate goals. In this case, a single call to sledgehammer on the statement can work where DSP fails.
>
> We have also included another baseline in Lean, which is ReProver, the retrieval-augmented model introduced in LeanDojo (https://arxiv.org/abs/2306.15626). Using the same search configuration (allocated 600 seconds), the search with and without the retrieval module is unable to solve any problem in Lean. We also performed a preliminary evaluation with an online proof-search model called Tactician (https://arxiv.org/abs/2008.00120) for Coq, yielding similar results. We will be including these experiments in a subsequent revision, and are happy to discuss more about this.

---

> > ### Author Rebuttal · Authors · 2024-08-15
> >
> > > miniF2F also provides some formal proofs if applicable. I am curious about the difficulty of manually proving the instances in PutnamBench. Could the easiest instance be proved manually in just a few steps?
> >
> > Indeed some of the easiest problems in our benchmark can be solved in proofs requiring < 10 lines. Some of the “easy” ones we are aware of: Putnam 1971 B1, 1986 B1, 1988 B1, 1988 B2,  2001 A1,  2012 A2. Some of these problems can admit short proofs which rely on some automated tactics in Lean, like nlinarith.
> >
> > > When using DSP or COPRA, one first needs to produce an informal proof. I am also curious about the informal proof accuracy by GPT-4. Can it correctly prove the easiest problem in PutnamBench?
> >
> > On the above problems we listed as being relatively easy, we did one query to GPT-4 with the informal problem statement. The result was 5 of 6 (informal) proofs being judged as correct by us. It is possible that this can be affected by direct contamination, or possibly the large amount of natural language reasoning data available to a model like GPT4. We also did the same with Putnam 2018 A5, 2020 B5, and 2021 A6, which are among the harder problems from recent years. On these problems, none of the generated proofs were accurate or sensible.
> >
> > We do want to underscore the value of a formal-theorem proving benchmark like PutnamBench. Especially in the case of evaluating new LLMs, it is useful (1) for producing precise reasoning paths which can be automatically checked, (2) demonstrating ability to operate in a low-resource language, (3) as it is free of direct contamination - the proofs to PutnamBench problems have for the most part never been written in any formal language and are not available online.
> >
> > To clarify, though COPRA does perform evaluations using informal proofs as an additional signal, our evaluation only uses the default configuration which does not incorporate informal proofs.
> >
> > >  For example, it seems unclear whether task 1 or task 2 is used in the evaluation.
> >
> > Thanks for mentioning this, all our evaluations are for task 2, where the problem statements have any additional data/numerical solutions written in. We will be sure to clarify this in the subsequent revision. To the best of our knowledge, the only existing system which supports Task 1 is Deepmind’s recent AlphaProof, which has yet only been announced in a blog post without further details.
> >
> > Thank you again for your detailed review, we would love to discuss further and address any further questions, comments, or suggestions you might have.

---

> > > ### Comment · Reviewer_pTSB · 2024-08-17
> > >
> > > Thank you for your detailed response. At this time, I have no further questions. I agree that PutnamBench would be a valuable benchmark for future neural theorem proving research. Therefore, I would like to maintain my original score.

---

> > > > ### Author Response · Authors · 2024-08-18
> > > >
> > > > Thank you again for the thorough review and consideration of our submission!

---

### Official Review · Reviewer_QCd7 · 2024-08-01
**Review of Submission 2041**

**Rating:** 7
**Confidence:** 4
**Correctness:** The claims are sound and correct.
**Clarity:** The paper is well-written and easy to…

**Review:**

The paper introduces a creative and innovative benchmark. While the evaluation of current SOTA LLMs) may be insufficient, the benchmark holds significant promise for advancing the field of automated theorem proving.

**Strengths:**

- The paper is well-motivated. It's good to see a benchmark bridging formal and informal theorem proving.
- The multilingal benchmark covers a wide range of undergraduate-level mathematical areas, addressing the limitations of previous datasets.

**Additional Feedback:**

See Opportunities For Improvement above.

**Documentation:**

The paper provides detailed documentation.

**Ethics:**

The paper has no ethical concerns.

**Limitations:**

The authors adequately addressed the limitations.

**Opportunities For Improvement:**

- The paper would benefit from additional details regarding the annotation process. Are the annotators students majoring in mathematics? How is semantic alignment ensured before and after formalization?
- Need to provide more baselines. What is the performance of language models at different scales, particularly those fine-tuned on formal mathematical libraries such as Mathlib?

**Relation To Prior Work:**

Relation to prior work is clearly discussed.

**Summary And Contributions:**

The paper presents PutnamBench, a multilingal evaluation benchmark for automated theorem proving. The benchmark consists of 1,337 human-annotated formalizations of 514 theorems from the William Lowell Putnam Mathematical Competition, encompassing a broad spectrum of undergraduate-level mathematics. Evaluation of both neural and symbolic approaches demonstrates the difficulty of the dataset.

---

> ### Author Rebuttal · Authors · 2024-08-15
>
> Thank you for the feedback and the kind words regarding our work! We respond to the main questions raised below. Please let us know if you have any other questions -- we would be happy to address them!
>
> > Are the annotators students majoring in mathematics?
>
> Most of our annotators are either mathematics degree holders or working towards an undergraduate degree in mathematics. In particular, the annotators consist of: 2 PhD students with undergraduate degrees in mathematics and with experience using interactive theorem provers; 5 Undergraduate students, of which 4 are majoring in math + CS and 1 is majoring in CS with a minor in philosophy. All students had prior experience to formal theorem proving beforehand and all are authors on the submission. We can include this information in a subsequent revision.
>
> > How is semantic alignment ensured before and after formalization?
>
> Semantic alignment does not have any automatic mechanism for verifying the accuracy of formalization, as the formalizations have to match a potentially noisy natural language specification. For example, the natural language problem statement might have some implicit hypotheses which have to be incorporated to produce a formalization.  The best way to ensure accuracy in the formalizations is to have multiple annotators look for mistakes. The annotation procedure was as follows: Given the natural language statement, one annotator was assigned to formalize the problem in Lean. A second (distinct) annotator did a review of the Lean formalization (while considering the natural language statement) and also produced a formalization in Isabelle. A third annotator then reviewed the Isabelle and Lean formalizations and produced the Coq formalization. Each formalization was reviewed by at least two different people. In this way, the number of semantic misalignments has been minimized and we expect the majority of problems to be correctly stated. We are also actively maintaining the benchmark and are readily receiving any feedback or requests from the community.
>
> > Need to provide more baselines. What is the performance of language models at different scales, particularly those fine-tuned on formal mathematical libraries such as Mathlib?
>
> We include below two more baselines. For Lean, we have included an evaluation of ReProver, the retrieval-augmented model used in LeanDojo (https://arxiv.org/abs/2306.15626). The model is a ~300m parameter ByT5 transformer finetuned on the LeanDojo training dataset, which consists of state-tactic pairs extracted from Mathlib. We ran proof search using the search configuration from that paper on each Lean formalization. We have also performed an experiment using Tactician (https://coq-tactician.github.io/api/introduction/), a method in Coq which uses locality sensitive hashing for online proof search. The Tactician evaluation is preliminary as we have so far ran with the configuration that does not incorporate background context. The results from the search are as follows:
>
> | Method                    | Successful Proofs |
> |---------------------------|-------------------|
> | ReProver w/ Retrieval      | 0/514                 |
> | ReProver w/o Retrieval     | 0/514                 |
> | Tactician                 | 0/309                 |
>
> We think that in the case of LeanDojo, it is not able to solve 1988 B1 as GPT-4 did because it requires an understanding that the choice $x=a-1,y=b-1,z=1$ eventually yields proper simplification of the resulting expression, which might not be possible with a model of its size. We will be sure to include these findings, and the full evaluation for Tactician, in the subsequent revision.
>
> At the time of submission, there were no larger models which were explicitly trained on formal math data that had better performance than calls to GPT-4 on similar tasks. Since GPT-4 was already quite incapable of solving more than a handful of problems in our benchmark, we thought it would not be worthwhile to benchmark smaller general-purpose language models. Recently there have been good models larger than ReProver which have finetuned on synthetic formal math data, like InternLM (https://github.com/InternLM/InternLM-Math) and DeepSeekProver (https://arxiv.org/abs/2405.14333). The former team has submitted results of their model to our leaderboard (https://trishullab.github.io/PutnamBench/leaderboard.html). The results show that smaller general-purpose models finetuned on math data, albeit with high compute requirements, can solve more problems than GPT4. We have previously contacted the other team to let them know about the benchmark and our leaderboard.
>
> We would also like to mention the difficulty of benchmarking approaches which go beyond simple LLM calls. These methods often have built custom implementations for interacting with the interactive theorem prover. To produce our evaluations we spent a significant amount of time and effort to modify the source code of various methods, which often do not work off-the-shelf. This can be because of version changes in the interactive theorem prover (or other dependencies) which are not supported by their custom tool. We did our best to include evaluations which cover the range of existing approaches.
>
> We would love to discuss further and address any further questions, comments, or suggestions you might have.

---

> > ### Comment · Reviewer_QCd7 · 2024-08-27
> >
> > Thank you to the authors for their efforts in the rebuttal. I acknowledge that adding additional baselines would require considerable effort to reproduce custom implementations, which is not necessary for this work. Therefore, I will maintain the original score.

---

### Official Review · Reviewer_DZY3 · 2024-08-13
**Great dataset/benchmark for the field**

**Rating:** 7
**Confidence:** 4
**Correctness:** Yes
**Clarity:** Yes

**Review:**

The dataset is novel, of high quality, and will certainly benefit the field overall.

**Strengths:**

- support for Coq/Rocq. The previous miniF2F fails to support Rocq effectively due to the dispersed math libraries. And I very much appreciate the authors’ efforts in constructing this dataset by consolidating several Rocq math libraries.
- factoring out solutions and trivial problems, which is clearly a big improvement over the previous miniF2F.
- well written paper with illustrative examples.

**Additional Feedback:**

N.A.

**Documentation:**

Yes

**Limitations:**

N.A.

**Opportunities For Improvement:**

- I would appreciate less demanding baselines like ReProver in LeanDojo and Proverbot9001 and Graph2Tac in Rocq. As GPT4 evolves and is relatively expensive, less demanding baselines based on open-source models could provide a more transparent comparison. This is of course too much to ask for during the rebuttal/discussion period  -- I just hope the authors can incorporate them in the future version of this paper.

**Relation To Prior Work:**

Yes, the different is well explained and the contributions are obvious.

**Summary And Contributions:**

This paper presents a novel formal math benchmark across multiple ITPs (Lean4, Isabelle, and Coq) as the previous miniF2F benchmark. The dataset is built upon the Putnam Mathematical Competition without any formal proofs, so that it nicely serves as an LLM reasoning benchmark without worrying too much about the data contamination problem. Neural and symbolic baselines like Sledgehammer, CORPRA, and DSP have been given for this new dataset.

---

> ### Author Rebuttal · Authors · 2024-08-15
>
> Thank you for your feedback and positive comments about our effort! We address the main points below.
>
> > I would appreciate less demanding baselines like ReProver in LeanDojo and Proverbot9001 and Graph2Tac in Rocq. As GPT4 evolves and is relatively expensive, less demanding baselines based on open-source models could provide a more transparent comparison.
>
> We agree! We also wanted to know the performance of a method which uses a neural model finetuned on formal proof data, so we have performed an evaluation using ReProver after our original submission. We did the evaluation using the default search configuration from the LeanDojo paper, with two experiments: one with and one without the retrieval model. We found that ReProver was unable to prove any of the Lean formalizations. In particular, it is unable to prove Putnam 1988 B1 which the other methods could, we expect this is because the particular choice `use a-1, b-1, 1` requires an understanding that the algebraic expression simplifies correctly later on. This understanding may not be as possible for a model of smaller size (ReProver is ~300m parameters).
>
> We considered including Proverbot, but it is trained on data from the CompCert verified C compiler project, which is in a different distribution than our competition math problems. We did not expect it to solve any problems for this reason, and hence did not include it in the evaluation.
>
> We considered including Graph2Tac, but their neural model was trained on data from Coq 8.11 and it is not clear how many of our formalizations, which are on Coq 8.18, are compatible. We did however recently benchmark Tactician (https://coq-tactician.github.io/api/introduction/), a method/framework similar to Graph2Tac (which Graph2Tac actually builds on) that uses locality sensitive hashing and works off-the-shelf for Coq 8.18. Our evaluation with Tactician is preliminary as so far we have only performed the experiment without including background context to the model. In this experiment, no Coq/Rocq problems were successfully proven. We will include the full evaluation in the subsequent revision.
>
> Please let us know if you have any other questions or comments, we would be happy to discuss!

---

> > ### Comment · Reviewer_DZY3 · 2024-08-27
> >
> > Thanks a lot for the detailed response. I look forward to the published version of this paper.

---

### Author Rebuttal · Authors · 2024-08-15

We thank all the reviewers for taking the time to assess our submission. We are grateful for the constructive feedback and the positive comments about the significance and novelty of our benchmark. Because the comments and questions raised have tended to be specific to each reviewer, we address them individually. Please let us know if you have more questions or would like to engage in further discussion!

---

### Decision · Program_Chairs · 2024-09-26

**Decision:**

Accept (Poster)

**Comment:**

This paper introduces PutnamBench, a new multilingual benchmark designed to evaluate neural theorem provers on formalized problems from the William Lowell Putnam Mathematical Competition. The dataset contains 1,337 formalizations of 514 theorems across Lean, Isabelle, and Coq. The problems span a wide range of undergraduate-level mathematics, making PutnamBench a challenging and comprehensive benchmark for assessing neural and symbolic theorem provers. The evaluation demonstrates that current state-of-the-art methods, including GPT-4, can solve only a small fraction of the problems, highlighting the difficulty and importance of this benchmark for advancing the field of formal reasoning.
After the rebuttal, the paper received strong support from reviewers (one "clear accept" and two "good paper, accept"). The primary concerns raised were the need for more baselines, details about the annotation process, and a more accurate description of the dataset’s multilingual nature. The authors addressed these issues in the rebuttal by adding evaluations for additional baselines, clarifying the qualifications of annotators, and discussing semantic alignment in formalization. The AC agrees with the reviewers' assessments and recommends accepting the paper.